# Crosstalk among Alternative Polyadenylation, Genetic Variants and Ubiquitin Modification Contribute to Lung Adenocarcinoma Risk

**DOI:** 10.3390/ijms25158084

**Published:** 2024-07-24

**Authors:** Yutong Wu, Yanqiong Yuan, Huiwen Xu, Wendi Zhang, Anhui Ning, Siqi Li, Qiong Chen, Xiaobo Tao, Gongbu Pan, Tian Tian, Lei Zhang, Minjie Chu, Jiahua Cui

**Affiliations:** Department of Epidemiology, School of Public Health, Nantong University, Nantong 226019, China; stoppeddeer@icloud.com (Y.W.); keleyyq@sina.com (Y.Y.); xuhuiwen@stmail.ntu.edu.cn (H.X.); zhangwendi@stmail.ntu.edu.cn (W.Z.); 13655515800@139.com (A.N.); 2317320009@stmail.ntu.edu.cn (S.L.); chenqiong99@stmail.ntu.edu.cn (Q.C.); txb791471146@163.com (X.T.); gongbu.pan@utas.edu.au (G.P.); ttyes_01@163.com (T.T.); zhanglei94@ntu.edu.cn (L.Z.)

**Keywords:** lung adenocarcinoma, ubiquitination, alternative polyadenylation, susceptibility

## Abstract

Ubiquitin modification and alternative polyadenylation play crucial roles in the onset and progression of cancer. Hence, this study aims to comprehensively and deeply understand gene regulation and associated biological processes in lung adenocarcinoma (LUAD) by integrating both mechanisms. Alternative polyadenylation (APA)-related E3 ubiquitin ligases in LUAD were identified through multiple databases, and the association between selected genetic loci influencing gene expression (apaQTL-SNPs) and LUAD risk were evaluated through the GWAS database of the Female Lung Cancer Consortium in Asia (FLCCA). Subsequently, the interaction between RNF213 and ZBTB20, as well as their functional mechanisms in LUAD, were investigated using bioinformatics analysis, Western blot, co-immunoprecipitation, and colony formation experiments. A total of five apaQTL-SNPs (rs41301932, rs4494603, rs9890400, rs56066320, and rs41301932), located on RNF213, were significantly associated with LUAD risk (*p* < 0.05), and they inhibit tumor growth through ubiquitin-mediated degradation of ZBTB20.

## 1. Introduction

Lung cancer remains the leading cause of death worldwide, causing approximately 2.2 million cases and 1.8 million deaths in 2020, with non-small-cell lung cancer (NSCLC) accounting for 85% [1,2]. According to pathological types, NSCLC can be divided into lung squamous-cell carcinoma (LUSC) and lung adenocarcinoma (LUAD). Of these, LUAD is the most common, accounting for 40% of lung cancers [3]. Despite diagnostic techniques and clinical treatment measures having improved in recent years, LUAD is mostly diagnosed at an advanced stage, resulting in a 5-year survival rate of only 5–17% [4]. LUAD remains a thorny issue in the malignancy field. Therefore, it is necessary to search for more new biomarkers of LUAD and explore its potential molecular mechanism to assist in identifying patients in the early stage and providing more therapeutic targets to improve patient survival.

The ubiquitin–proteasome system (UPS) is the main pathway of protein degradation and regulates most cellular processes [5]. Protein degradation by UPS is a multi-stage process that requires the participation of a variety of enzymes, among which the most critical enzymes include ubiquitin-activating enzymes (E1s), ubiquitin-binding enzymes (E2s), and ubiquitin-protein ligases (E3s) [6]. E1s activate ubiquitin molecules and send them to E2s, which bind ubiquitin molecules to the proteins to be degraded, while E3s identify the proteins to be degraded [7]. Increasing studies have shown that E3s influence tumor development by ubiquitinating and degrading many transcription factors associated with human cancer when they are functionally disabled or improperly targeted [8]. For example, TRIM46 degrades PHLPP2 through ubiquitination to activate AKT/HK2 signaling and thus promote LUAD cell proliferation [9]. Wang et al. found that the AMPK–HOXB9–KRAS axis can influence the occurrence and development of LUAD [10], thus finding out E3s related to cancer may help to explain the potential mechanism of tumorigenesis and progression.

Ring finger protein 213 (RNF213), also known as mysterin, is the largest E3 ubiquitin ligase known to humans, with a molecular weight of 591 kDa [11]. The AAA+ domain and Ring domain in its structure give it ATPase activity and E3 ubiquitin ligase action [12]. RNF213 belongs to the RNF subfamily of the E3 ubiquitin ligase superfamily. Numerous studies have revealed a close association between members of the RNF family and cancer [13]. In LUAD, the RNF family also plays a pivotal role; for instance, RNF115 promotes LUAD by mediating APC ubiquitination and activating the Wnt/β-catenin pathway [14]. At present, it is clear that the mutant of RNF213 is a susceptibility gene for Moyamoya disease, and RNF213 also plays a dangerous role in other cerebrovascular diseases such as intracranial aortic stenosis/occlusion and intracranial aneurysms [15]. In addition, Li et al. have found that RNF213 can be used as a driver gene and prognostic marker of hepatocellular carcinoma [16]. Existing studies on the ubiquitination of RNF213 primarily focus on its role in the ubiquitination of lipopolysaccharides during bacterial infection [12] and its function as an E3 ligase in degrading RTA through the ubiquitin–proteasome pathway, which inhibits both de novo infection and lytic reactivation of gamma herpesvirus [17]. However, the role of RNF213 as an E3 ubiquitin ligase in LUAD is rarely discussed.

E3s play a crucial role in regulating the ubiquitination and degradation of various proteins in tumors, but few studies have reported why E3s are expressed abnormally during tumor progression. This is a question worthy of concern and discussion. Alternative polyadenylation (APA) is an RNA processing event that produces 3′untranslated regions (3′UTR) of varying length by recognizing different polyadenylation signals (PAS) during the maturation of precursor mRNA, affecting mRNA stability, localization, transport, and translation [18]. APA events are widespread in eukaryotes, more than 70% of human protein-coding genes encode multiple polyadenylation sites (PAS) [19]. Choosing the proximal PAS for splicing means a shorter 3′UTR, while choosing the distal PAS means a longer 3′UTR. At present, many studies have shown that the disorder of APA may be closely related to the abnormal expression of oncogenes and oncosuppressor genes, including E3s. For example, the oncosuppressor gene XBP1, a short 3′UTR inhibited p53 ubiquitination compared to a long 3′UTR, promoting LUAD cell repair [20]. Here, we elucidate whether APA events influence the occurrence and progression of LUAD through protein ubiquitination.

In this study, we conducted a comprehensive analysis across multiple databases and identified RNF213 as a carcinogenic factor in LUAD. Subsequently, leveraging APA-related databases and GWAS analysis, we determined genetic loci influencing RNF213 expression (apaQTL-SNPs) and elucidated how mutations at these loci affect LUAD risk by modulating RNF213 expression. Through cell experiments, we confirmed that silencing RNF213 expression inhibit LUAD cell proliferation. Furthermore, we clarified that RNF213 mediates ubiquitination and degradation of ZBTB20 in LUAD cells, further promoting carcinogenesis. Our findings may offer new strategies for the diagnosis and treatment of LUAD, with RNF213 potentially serving as a novel biomarker for LUAD.

## 2. Results

### 2.1. Screening for APA-Related E3s in LUAD

To explore whether APA events influence the occurrence and progression of LUAD through protein ubiquitination, we combined multiple databases to screen APA-related E3s. As shown in Figure 1, in the first section, we screened for differentially expressed proteins/genes in LUAD and adjacent non-tumors. Specifically, we compared the expression of various proteins in LUAD tumors and paired adjacent non-tumors in the CPTAC LUAD Discovery study and the Chinese LUAD study, respectively, and collectively identified 3733 proteins with consistent expression differences (*P*_FDR_ < 0.05). With the same process as above, LUAD-related genes with different expression in tumor and normal tissues were screened out, and 6452 genes were identified as having high or low expression in both TCGA-LUAD and the Chinese study.

In the second section, we consolidated the protein and gene results from the first part, identifying 1420 genes and proteins with consistent differential expression. Specifically, we used fold change (FC) to assess the consistency between protein and mRNA expression. Genes/proteins with log_2_FC > 0 were defined as upregulated, while those with log_2_FC < 0 were defined as downregulated, identifying 1420 genes and proteins with consistent differential expression (Appendix A). Additionally, by combining this with 414 E3s extracted from human sapiens, we identified a total of 19 E3s associated with LUAD.

Next, Cui et al. [21] conducted a study collecting information on SNP sites with APA functionality, providing 1.49 million APA-functional SNP sites located in the 3′UTR. We identified a total of 473,008 apaQTL-SNPs regulating 5888 genes in human lung tissue, and further filtered apaQTL-SNPs associated with LUAD-related E3s. Using a minor allele frequency (MAF) > 0.05 in the Chinese Han population and an *r*^2^ threshold < 0.8 for linkage disequilibrium (LD) analysis, we narrowed down the apaQTL-SNPs to 49 potentially correlated with LUAD risk (Appendix A).

Finally, to determine the relationship between apaQTL-SNPs screened by the above steps and the risk of LUAD, we performed a logistic regression analysis using age as a covariable through the GWAS database of the Female Lung Cancer Consortium in Asia (FLCCA). The FLCCA pooled data from five studies in China (Taiwan and Hong Kong), South Korea, Japan and Singapore involving 3453 LUAD patients and 3710 healthy controls [22]. The FLCCA GWAS analyzed 49 candidate apaQTL-SNPs located in HECW2, TRIM24, PDLIM2 and RNF213, and finally 5 significant apaQTL-SNPs all located in RNF213 were obtained.

### 2.2. RNF213 Was Positively Correlated with the Poor Prognosis of LUAD

In terms of proteomics, the expression of RNF213 in LUAD tumors was higher than that in normal tissues between 101 pairs of the CPTAC LUAD Discovery study (*p* = 2.56 × 10^−16^) and 103 pairs of the Chinese-LUAD study (*p* = 3.57 × 10^−33^) (Figure 2A,B). In accordance with clinical information, the expression of RNF213 was analyzed in groups according to clinical stage and metastasis. The results showed that there were differences in the protein expression of RNF213 in different clinical stages (*p* = 0.011) and pathological N (*p* = 0.02) in the Chinese-LUAD study data, and the higher the level of clinical stage, the wider the scope of tumor metastasis, corresponding the higher expression of RNF213 (Figure 2C,D). The survival analysis results showed that individuals with low RNF213 protein expression had a higher survival probability than those with high RNF213 expression (*p* = 0.0013) (Figure 2E). In addition, survival results combined with clinical information showed that the probability of survival was higher when RNF213 was low in expression and had a lower clinical stage or less metastasis (Figure 2F–H). This evidence suggests that high expression of RNF213 may be a risk factor for LUAD.

In terms of gene expression, based on the TCGA-LUAD database, RNF213 gene expression was significantly higher in LUAD tumors (*n* = 513) than in normal controls (*n* = 59) (*p* = 1.42 × 10^−6^) (Figure 2I). RNF213 gene expression in 51 tumors in the Chinese-LUAD data was also significantly higher than that in 49 normal controls (*p* = 2.80 × 10^−8^) (Figure 2J).

### 2.3. Mutations in apaQTL-SNPs Regulating RNF213 Are Associated with an Increased Risk of LUAD

The association between apaQTL-SNPs and LUAD risk was evaluated using the FLCCA GWAS database. The results showed that 5 apaQTL-SNPs located at 1MB of RNF213 (rs41301932, rs4494603, rs9890400, rs56066320, and rs13341123) was significantly associated with increased risk of LUAD (additive model: OR > 1, *p* < 0.05) (Table 1).

Percentage of Distal poly(A) site Usage Index (PDUI) is a DaPars algorithm used to quantify APA events. The value of PDUI ranges from 0 to 1. If PDUI is close to 1, the gene tends to use a longer 3′UTR, and conversely the gene carries a shorter 3′UTR. We further assessed the relationship between the identified apaQTL-SNPs in the lungs and the PDUI values of RNF213. As shown in Figure 3, all five apaQTL-SNPs exhibited a significant correlation with the PDUI values of RNF213. Compared to the wild-type homozygous genotype, both heterozygous and mutant homozygous genotypes showed lower PDUI values.

### 2.4. ZBTB20 Is a Potential Substrate for Ubiquitination by RNF213 in LUAD

A total of 23 predicted substrates for RNF213 were downloaded through Ubi Browser 2.0, of which GLUD1 is the reported known substrate of RNF213. Since RNF213 is highly expressed in cancer, its ubiquitination substrate should theoretically be low expressed in cancer. Therefore, we selected proteins that were significantly lower expressed in cancer both in the CPTAC Discovery study and the Chinese-LUAD study, and found that two of the predicted substrates, namely ZBTB20 and GLUD1 (Figure 4A), were lower in cancer (Figure 4B–E). Survival analysis showed an inconclusive association between GLUD1 and LUAD survival (*p* = 0.088) (Figure 4F), while high expression of ZBTB20 was associated with greater survival (*p* = 0.028) (Figure 4G). Further exploration revealed that survival of clinical stage and metastasis of ZBTB20 was also statistically significant (Figure 4H,I). Low expression of ZBTB20 and more advanced clinical stage or metastasis mean a lower probability of survival.

### 2.5. RNF213 Mediates ZBTB20 Ubiquitination and Degradation to Suppress LUAD Progression

The Protein expression levels of RNF213 and ZBTB20 in LUAD were evaluated by The Human Protein Atlas database. As shown in Figure 5A,B, the immunostaining intensity of RNF213 was concentrated in strong (*n* = 1), moderate (*n* = 4) and weak (*n* = 5), while ZBTB20 was concentrated in moderate (*n* = 1), weak (*n* = 2) and negative (*n* = 7). These results indicate that RNF213 is highly expressed in LUAD, while ZBTB20 is conversely low expressed in LUAD.

In order to verify the interaction between RNF213 and ZBTB20, firstly, we conducted a correlation analysis through public databases. In both the CPTAC Discovery study (Rs = −0.36, *p* = 1.87 × 10^−6^) (Figure 5C) and the Chinese LUAD study (Rs = −0.56, *p* = 4.64 × 10^−18^) (Figure 5D), ZBTB20 exhibited a negative correlation with RNF213. Figure 5E depicts the molecular docking model illustrating the interaction between RNF213 and ZBTB20 (docking score = −296.18; confidence score = 0.949). Subsequently, the interactions between RNF213 and ZBTB20 were also confirmed by Co-IP assays in SPCA1 and PC9 cells (Figure 6A).

To explore the role of RNF213 in the ubiquitination of ZBTB20 and its biological function in LUAD, we successfully transfected LUAD cell lines SPCA1 and PC9 with siRNA-mediated RNF213 knockdown (Figure 6B). Treatment with the protease inhibitor MG132 partially blocked the degradation of ZBTB20 (Figure 6C). Additionally, we used ZBTB20^high^ plasmids, RNF213^low^, and combined ZBTB20^high^ plasmids and RNF213^low^ treatments in SPCA1 and PC9 cells (Figure 6D) and evaluated their effects on the proliferation of LUAD cell lines using the CCK8 assay. The results showed that the proliferation ability of SPCA1 and PC9 cells in the RNF213 knockout group was significantly reduced compared to the wild-type group (*p* < 0.0001). Furthermore, the combined use of si-RNF213 and Flag-ZBTB20 further decreased cell proliferation ability (*p* < 0.0001) (Figure 6E,F). In the SPCA1 cell line, we also obtained similar results through colony formation assays (Appendix A). In summary, RNF213 may mediate the degradation of ZBTB20 through the ubiquitin–proteasome pathway and is associated with oncogenic effects in LUAD tumorigenesis.

## 3. Discussion

In this study, we delineate the role of RNF213 in the progression of LUAD. Both at the protein and gene levels, RNF213 exhibits elevated expression in LUAD tissues, correlating with a poorer prognosis. Leveraging FLCCA GWAS, we systematically assessed the relationship between apaQTL-SNPs and LUAD risk. The results reveal that mutations in five cis-apaQTL-SNPs (rs41301932, rs4494603, rs9890400, rs56066320, and rs3341123) are significantly associated with increased RNF213 expression and elevated LUAD risk. To further validate the role of RNF213 in LUAD, we transfected plasmids silencing RNF213 expression. CCK8 assay demonstrate that silencing RNF213 expression inhibits LUAD cell proliferation. Additionally, the heightened expression of the predicted ubiquitination substrate, ZBTB20, also exhibits inhibitory effects on LUAD. Collectively, this evidence suggests that RNF213 has the potential to serve as a novel biomarker for LUAD.

ZBTB20 (Zinc Finger and BTB Domain Containing 20) is comprised of zinc finger structures and a BTB domain (Broad-Complex, Tramtrack, and Bric-a-Brac domain), with the BTB domain responsible for protein-protein interactions [23]. As of current knowledge, ZBTB20 is essential for normal glucose homeostasis and serves as a key regulatory factor for lipid homeostasis [24]. Despite some studies indicating an association between ZBTB20 and certain cancers such as liver cancer [25] and gastric cancer [26], its connection with LUAD and the underlying mechanisms remain largely unknown. In our study, ZBTB20 was found to be downregulated in LUAD tissues, correlating with a better prognosis. Overexpression of ZBTB20 inhibited the progression of LUAD cells, specifically SPCA1. Moreover, ZBTB20 may be subject to ubiquitination by RNF213, leading to a reduction in ZBTB20 expression and consequently promoting LUAD. This discovery suggests that ZBTB20 could potentially function as a suppressor in LUAD.

Previous research has primarily focused on revealing the association between RNF213 and vascular-related diseases [27,28]. Mutations in the RNF213 gene have been identified in relation to an increased risk of Moyamoya disease and aneurysm development [29]. However, the direct relationship between RNF213 and cancer has not been extensively explored. In this study, we elucidated the potential mechanisms through which RNF213 influences the occurrence of LUAD. Specifically, we discovered that five apaQTL-SNPs (rs41301932, rs4494603, rs9890400, rs56066320, and rs3341123) within a 1 Mb range of RNF213 regulate its expression. We hypothesize that mutations in these apaQTL-SNPs lead to a decrease in RNF213’s PDUI value, suggesting the selection of a proximal PAS and resulting in a shorter 3′UTR. The 3′UTR contains numerous regulatory elements, including miRNA binding sites. A shorter 3′UTR allows for miRNA escape, leading to an elevated expression level of RNF213 mRNA [30,31]. Subsequently, RNF213, through its ubiquitination pathway, degrades more ZBTB20, thereby increasing the risk of LUAD in the population.

This study aims to provide a more comprehensive and in-depth understanding of gene regulation and related biological processes by integrating ubiquitination and alternative polyadenylation (APA) in the analysis of RNF213. The advantages include, but are not limited to: (1) Comprehensive gene expression regulation information: Ubiquitination and APA represent two distinct regulatory mechanisms. Through integrated analysis, both post-transcriptional and post-translational regulation of genes can be considered, contributing to a more comprehensive understanding of the regulatory network of gene expression. (2) Discovery of novel functional associations: Integrated analysis aids in uncovering functional associations between alternative polyadenylation and ubiquitination. Certain variants in alternative polyadenylation may be associated with ubiquitin ligase targets, influencing cellular signaling pathways and biological processes. (3) More comprehensive understanding of disease relevance: By simultaneously considering APA and ubiquitination information, a more comprehensive understanding of their roles in diseases can be achieved. This contributes to the identification of potential biomarkers, therapeutic targets, and insights into disease mechanisms.

Collectively, our study identified genetic loci influencing RNF213 expression (apaQTL-SNPs) and established the association between these mutations and RNF213 expression, as well as their impact on LUAD risk. Subsequently, RNF213 mediates ubiquitination and degradation of ZBTB20 in LUAD cells, further promoting carcinogenesis (Figure 7). This offers new insights into the hierarchical structure of the transcriptional and translational regulatory network contributing to the occurrence of LUAD.

Although this study identified five apaQTL-SNPs (rs41301932, rs4494603, rs9890400, rss56066320, and rs41301932) on RNF213 that may influence the development of LUAD through ubiquitin-mediated ZBTB20, there are still some limitations. The study primarily relied on analysis of public databases and querying public datasets for scientific work. Limitations of this approach include the inability to establish causal relationships and reliance on existing data, which may lack the depth and specificity needed for comprehensive mechanistic insights. Therefore, future research should employ more complex experimental models to validate these findings and confirm the biological relevance and mechanistic pathways involved.

## 4. Materials and Methods

### 4.1. Acquisition and Processing of Proteomics and Genomics Data

The LUAD proteomics data were derived from two released studies published in cell, the CPTAC LUAD Discovery study by Gillette et al. [32] and the Chinese LUAD study by Xu et al. [33]. The CPTAC LUAD Discovery study used Tandem mass tags (TMT) technology to measure the relative protein content of 110 LUAD tumor samples and 101 pairs of adjacent non-tumor tissues, and log the TMT ratio. Then, 101 pairs of matched samples were selected for the follow-up study. 103 LUAD tumors and their paired non-tumors from the Chinese study were used a marker-free quantitative algorithm to quantify proteins and quantile normalization to process the data. In this study, we recorded the values after scaling the data by mean and median. Horizontal standardization uses the mean to align data distribution characteristics of the same protein between different individuals, enhancing comparability. Then, vertical standardization calculates the median protein levels within the same treatment group to normalize different proteins, ensuring data comparison on the same scale within each individual.

Gene expression data were sourced from two avenues—firstly, from the study by Xu et al. [33], which provided gene data for Chinese LUAD, comprising 51 tumor samples and 49 non-tumor tissue samples. Additionally, data was obtained from GDC TCGA-LUAD on UCSC Xena (https://xenabrowser.net/, accessed on 8 May 2023), where, after removing duplicate data from the same individual, 513 tumor samples and 59 adjacent non-tumor samples were retained. To maintain consistency, both TCGA-LUAD gene expression data and Chinese LUAD gene data were processed using log2(FPKM + 1). Fold change (FC) is used to quantify the relative changes in gene or protein expression levels between tumor and healthy conditions. Detailed clinical data of the above cohort are shown in Table 2.

### 4.2. Prediction of Substrate Proteins

The predicted substrates of the E3s identified in the above steps can be found on Ubi Browser 2.0 (http://ubibrowser.bio-it.cn/ubibrowser_v3/home/index, accessed on 5 May 2023). Protein expression data from CPTAC LUAD Discovery research and Chinese LUAD proteomics studies can be employed for further substrate selection.

### 4.3. Protein-Protein Docking Assay

The 3D protein structure of RNF213 (PBD ID: 7OIK) was obtained from the RCSB PDB (https://www.rcsb.org/, accessed on 16 May 2023) database, and the 3D structure of ZBTB20 was obtained from the AlphaFold Protein Structure Database (https://alphafold.com/, accessed on 16 May 2023). The HDOCK server (http://hdock.phys.hust.edu.cn/, accessed on 16 May 2023) was used for docking protein-protein assay. The server automatically predicts the interactions between them based on a hybrid algorithm of templated and template-free docking [34], scores the protein-protein docking model via the function ITScorePP and empirically defines a confidence score associated with the docking score that indicates the likelihood of the two molecules binding. A confidence score higher than 0.7 indicates that two molecules are likely to combine.

### 4.4. Cell Culture and Transfection

The human LUAD cell lines SPCA1 and PC9 was purchased from ATCC (Gaithersburg, MD, USA) and maintain in DMEM with 10% fetal bovine serum, 100 U/mL penicillin, and 100 μg/mL streptomycin at 37 °C with 5% CO_2_. Acquiring commercially available RNF213 and ZBTB20 plasmids, with the design and synthesis of si-RNF213 and Flag-ZBTB20 performed by Nanjing Coreis Biotech Co., Ltd., Nanjing, China. DNA plasmids or siRNA were transfected into SPCA1 cells using Lipo8000^TM^ Transfection Reagent (Beyotime, Shanghai, China) according to the manufacturer’s instructions. The successful transfection of plasmid was determined by Western blot assay.

### 4.5. Western Blot Assay

The cells were lysed with RIPA lysis buffer (Beyotime) and protease inhibitor (Roche, Shanghai, China) to extract total protein after washing with pre-cooled PBS. The protein concentration was detected by the protein quantitative BCA (Bicinchoninic acid) method, and the corresponding samples were prepared. Subsequently, protein separation was performed using 10% sodium dodecyl sulfate-polyacrylamide gel electrophoresis (SDS-PAGE) and transferred to a polyvinylidene fluoride (PVDF) membrane (Millipore, Billerica, MA, USA). The membrane was blocked with 5% skimmed milk powder and incubated overnight at 4 °C with antibodies against RNF213 (1:1000, Santa Cruz Biotechnology, Dallas, TX, USA), ZBTB20 (1:2000, Proteintech, Wuhan, China), and Tubulin (1:1000, Beyotime). Finally, the membrane was incubated with goat anti-mouse secondary antibody (1:1000, Beyotime) and goat anti-rabbit secondary antibody (1:1000, Beyotime) as per the specifications, followed by color development. The relative expressions of target proteins were standardized against Tubulin, and relative quantification of immunoblot images was performed using ImageJ software (version 1.54d).

### 4.6. Co-Immunoprecipitation (Co-IP) and Ubiquitination Assay

For Co-immunoprecipitation (Co-IP), total protein was extracted from SPCA1 cells, as described in the Section 4.5. Following the acquisition of cell lysates, they were incubated overnight at 4 °C with either specific antibodies or control IgG (Santa Cruz Biotechnology, Shanghai, China). Then, Protein A/G agarose beads (Santa Cruz Biotechnology) were added and incubated at 4 °C for 2 h to capture the antigen-antibody complexes. The agarose bead-antibody-protein complexes were then centrifuged, washed four times, and dissolved in 60–80 μL of 2× loading buffer for Co-IP detection. For ubiquitination assays, si-RNF213 and His-Ub were co-transfected into SPCA1 cells. Prior to cell lysis, the cells were treated with the proteasome inhibitor MG132 (10 μM) for an additional 6 h. Subsequently, Western blot analysis was performed for evaluation

### 4.7. Cell Counting Kit-8 (CCK8) Assay

The proliferation of SPCA1 and PC9 cells was assessed using the CCK8 assay (Beyotime). Specifically, an appropriate number of SPCA1 and PC9 cells (2000 cells/100 μL) were seeded into 96-well plates, with approximately 100 μL of cell suspension per well. According to the manufacturer’s protocol for the CCK8 assay, 10 μL of CCK8 solution (10% CCK8 in culture medium) was added to each well at 0, 24, 48, and 72 h of culture. After incubating at 37 °C for 1–4 h, absorbance was measured at 450 nm. Each plate included a blank control, and three replicates were set up for statistical analysis.

### 4.8. Statistical Analysis

The “limma” package in R was utilized to analyze protein and gene expression differences between LUAD tumors and healthy controls. One-way ANOVA was employed to assess differences in protein expression among three or more groups. Logistic regression analysis was used for the association between these candidate apaQTL-SNPs and the risk of LUAD after adjusting age, and odds ratio (ORs) and 95% confidence intervals (95%CI) were used as indicators. The correlation strength between E3s and their predicted substrates was evaluated using Pearson’s correlation coefficient (R), and the relationship was visualized using the “ggstatsplot” package in R. Kaplan–Meier survival curves were generated using the “survival” package. Differences were considered statistically significant at *p* < 0.05. Statistical analyses were performed using R version 4.2.1 (R Foundation for Statistical Computing, Vienna, Austria) and Stata version 15.0 (StataCorp, College Station, TX, USA).

## Figures and Tables

**Figure 1 ijms-25-08084-f001:**
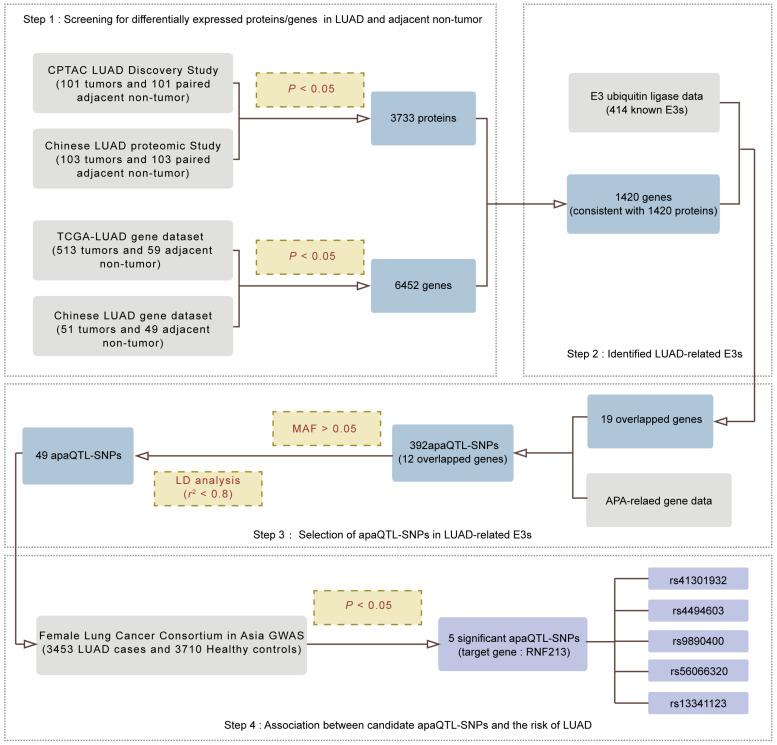
Flowchart of RNF213 screening. apaQTL, alternative polyadenylation quantitative trait loci; LUAD, lung adenocarcinoma; MAF, minor allele frequency; LD, linkage disequilibrium.

**Figure 2 ijms-25-08084-f002:**
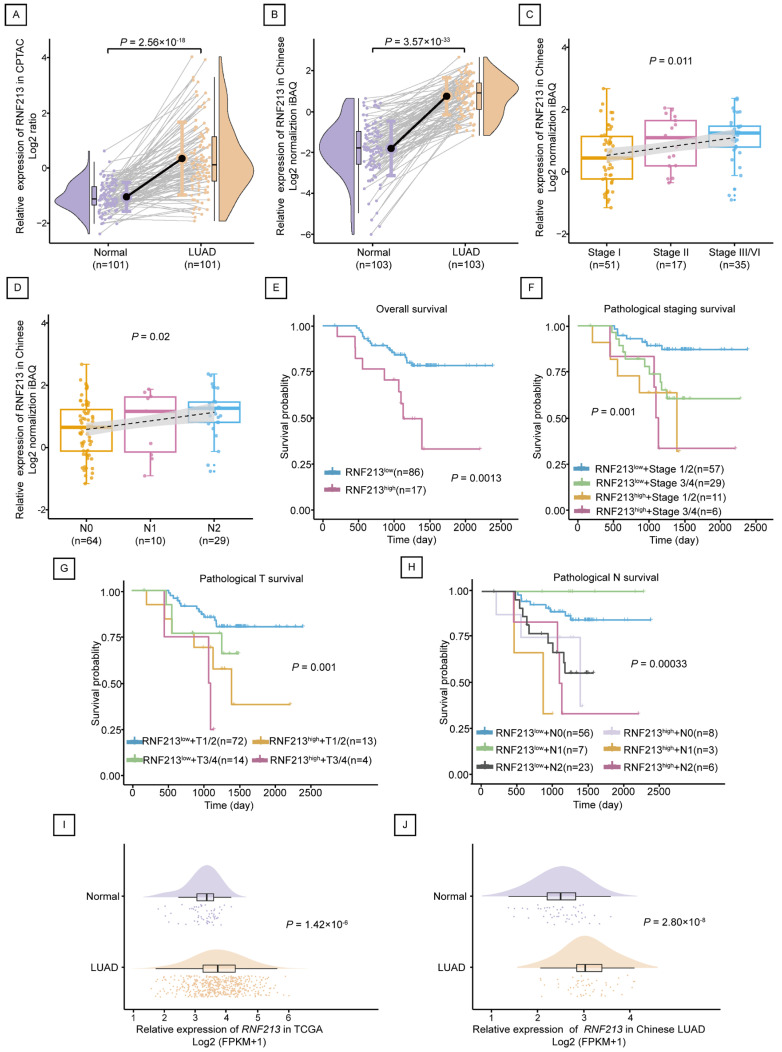
RNF213 was upregulated in LUAD associated with poor prognosis. (**A**,**B**) Protein expression of RNF213 in 101 pairs of LUAD tissues and adjacent normal tissues from the CPTAC database (**A**) and 103 pairs of LUAD tissues and adjacent normal tissues from the Chinese LUAD database (**B**), respectively. (**C**,**D**) Protein expression of RNF213 based on pathological staging (**C**) and pathological N (**D**) in the Chinese LUAD database. The gray area represents the 95% confidence interval (CI). (**E**–**H**) Overall survival curve of RNF213 in the Chinese LUAD database (**E**), including pathological staging (**F**), pathological T (**G**), and pathological N (**H**). (**I**,**J**) mRNA expression of RNF213 in TCGA (513 LUAD and 59 normal tissues) (**I**) and Chinese LUAD database (51 LUAD and 49 normal tissues) (**J**), respectively.

**Figure 3 ijms-25-08084-f003:**
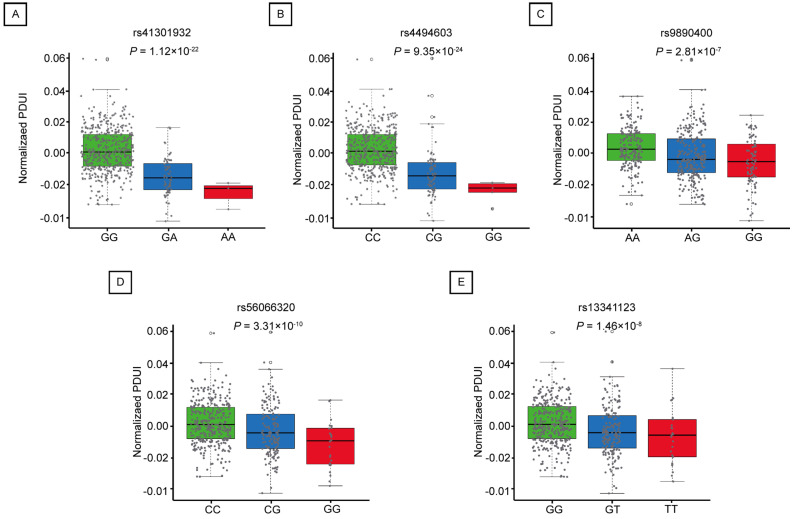
The relationship between five candidate apaQTL-SNPs and PDUI values of RNF213. (**A**–**E**) rs41301932 (**A**); rs4494603 (**B**); rs9890400 (**C**); rs56066320 (**D**); rs13341123 (**E**).

**Figure 4 ijms-25-08084-f004:**
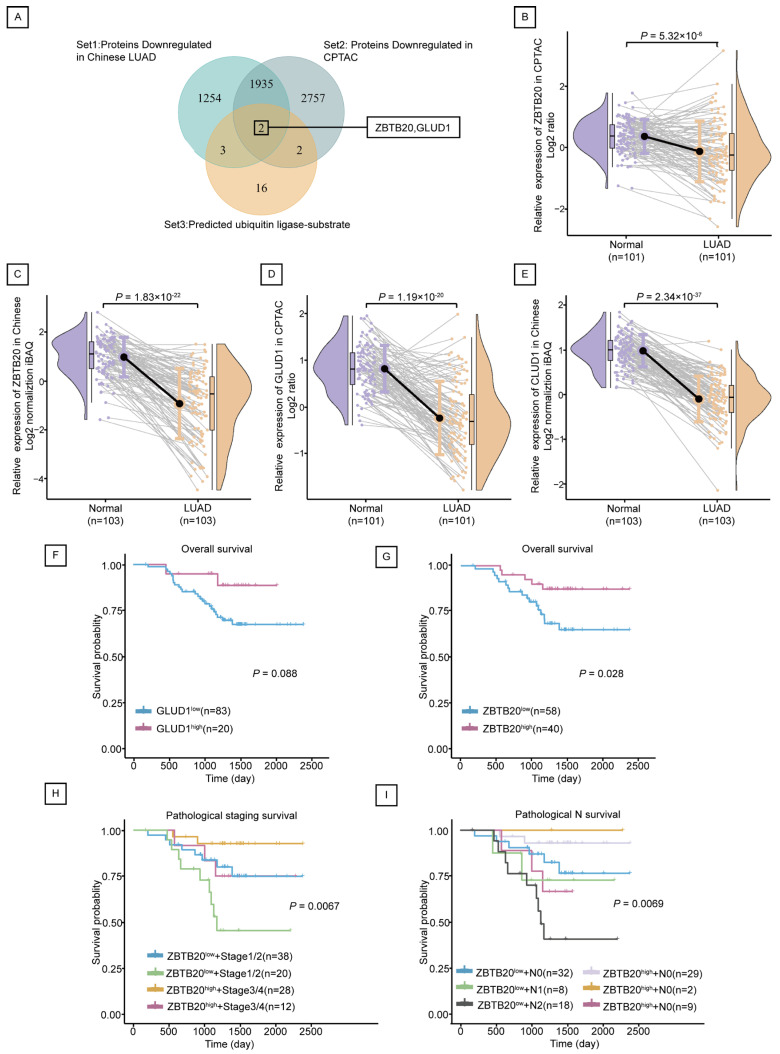
ZBTB20 was downregulated in LUAD associated with a favorable prognosis. (**A**) Venn diagram of RNF213 ubiquitination substrates screening in LUAD. (**B**,**C**) Protein expression of ZBTB20 in CPTAC database (101 pairs of LUAD tissues and adjacent normal tissues) (**B**) and Chinese LUAD database (103 pairs of LUAD tissues and adjacent normal tissues) (**C**), respectively. (**D**,**E**) Protein expression of GLUD1 in CPTAC database (101 pairs of LUAD tissues and adjacent normal tissues) (**D**) and Chinese LUAD database (103 pairs of LUAD tissues and adjacent normal tissues) (**E**), respectively. (**F**–**I**) Overall survival curve of GLUD1 (**F**) and ZBTB20 (**G**) in the Chinese LUAD database. Survival curves of ZBTB20 stratified by pathological staging (**H**) and pathological N (**I**), respectively.

**Figure 5 ijms-25-08084-f005:**
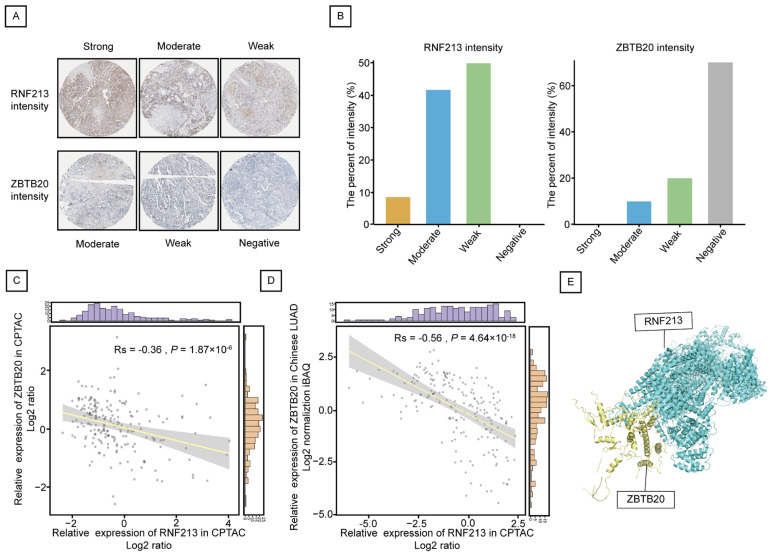
ZBTB20 is negatively correlated with RNF213 in LUAD. (**A**) The representative immunohistochemical staining intensity images of RNF213 and ZBTB20 based on the database of The Human Protein Atlas. (**B**) displays the percentage of immunohistochemical staining categorized as strong, moderate, weak, and negative intensity in LUAD patients. RNF213 is represented on the left, while ZBTB20 is shown on the right. (**C**,**D**) The correlation of RNF213 and ZBTB20 expression was demonstrated by CPTAC database (**C**) and the Chinese LUAD database (**D**). (**E**) The molecular docking model between RNF213 and ZBTB20 was presented by the HDCOK server.

**Figure 6 ijms-25-08084-f006:**
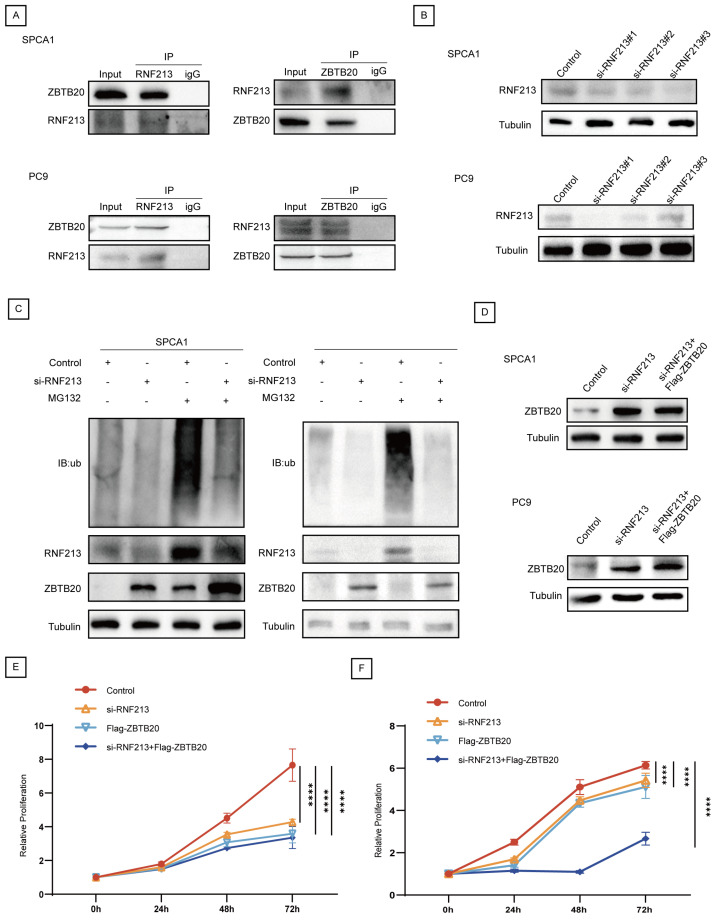
ZBTB20 ubiquitination by RNF213 in LUAD. (**A**) Co-immunoprecipitation (Co-IP) detecting the interaction between RNF213 and ZBTB20. (**B**) SPCA1 and PC9 were transfected with siRNF213 and subject to Western blot analysis. (**C**) Ubiquitination in SPCA1 and PC9 cells transfected with MG132, siRNF213 was determined by Western blot analysis. (**D**) The expression level of ZBTB20 in SPCA1 and PC9 cells in indicated groups was analyzed by Western blot. (**E**) The CCK8 assay in SPCA1 cells. (**F**) The CCK8 assay in PC9 cells (**** *p* value < 0.0001).

**Figure 7 ijms-25-08084-f007:**
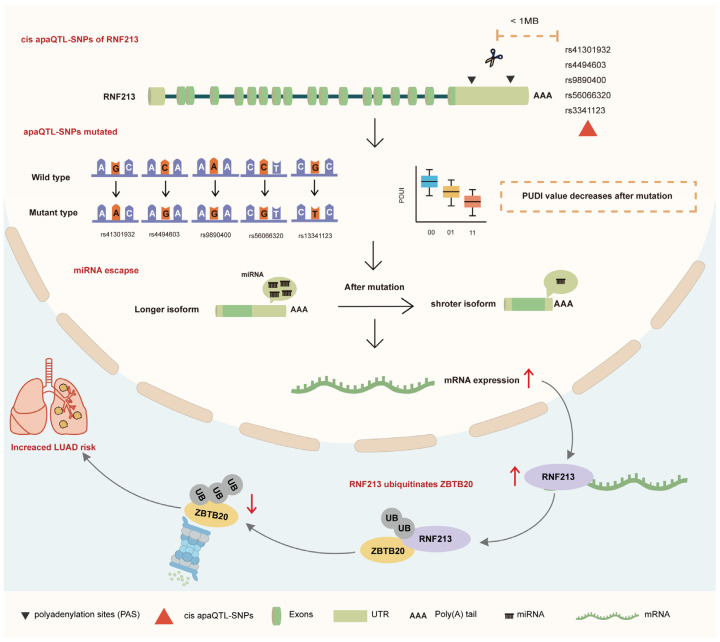
The potential mechanism of apaQTL-SNPS mutations regulating RNF213 in affecting the Progression of LUAD.

**Table 1 ijms-25-08084-t001:** Characteristics of the 6 cis apaQTL-SNPs in FLCCA GWAS.

SNPs	Gene	Location (hg37)	Genotypes	Cases, *n* (100%)	Controls, *n* (100%)	Adjusted OR (95%CI) ^a^	*p*
rs41301932	*RNF213*	chr17:78406974	GG	3109 (90.04%)	3413 (92.00%)	1 (ref)	-
			GA	333 (9.64%)	291 (7.84%)	1.26 (1.07–1.49)	0.005
			AA	11 (0.32%)	6 (0.16%)	2.05 (0.76–5.57)	0.157
			Dominant model			1.28 (1.09–1.51)	0.003
			Additive model			1.28 (1.09–1.50)	0.002
rs4494603	*RNF213*	chr17:78408213	CC	3056 (88.50%)	3373 (90.92%)	1 (ref)	-
			CG	385 (11.15%)	327 (8.81%)	1.31 (1.12–1.53)	0.001
			GG	12 (0.35%)	10 (0.27%)	1.32 (0.57–3.07)	0.516
			Dominant model			1.31 (1.12–1.52)	0.001
			Additive model			1.29 (1.11–1.49)	0.001
rs9890400	*RNF213*	chr17:78416493	AA	1097 (31.77%)	1267 (34.15%)	1 (ref)	-
			AG	1641 (47.52%)	1732 (46.68%)	1.09 (0.98–1.21)	0.106
			GG	715 (20.71%)	711 (19.16%)	1.16 (1.02–1.32)	0.028
			Dominant model			1.11 (1.01–1.23)	0.037
			Additive model			1.08 (1.01–1.15)	0.023
rs56066320	*RNF213*	chr17:78418906	CC	2942 (85.20%)	3257 (87.79%)	1 (ref)	-
			CG	493 (14.28%)	437 (11.78%)	1.25 (1.09–1.44)	0.002
			GG	18 (0.52%)	16 (0.43%)	1.30 (0.66–2.55)	0.453
			Dominant model			1.25 (1.09–1.44)	0.001
			Additive model			1.23 (1.08–1.40)	0.001
rs13341123	*RNF213*	chr17:78427033	GG	2987 (86.50%)	3297 (88.87%)	1 (ref)	-
			GT	451 (13.06%)	400 (10.78%)	1.24 (1.07–1.43)	0.003
			TT	15 (0.44%)	13 (0.35%)	1.14 (0.79–1.66)	0.486
			Dominant model			1.24 (1.08–1.43)	0.003
			Additive model			1.23 (1.07–1.41)	0.003

^a^ Logistic regression analysis adjusted for age.

**Table 2 ijms-25-08084-t002:** Characteristics of the subjects enrolled in this study.

Variables	CPTAC Discovery Study		Chinese LUAD Study		TCGA-LUAD		FLCCA GWAS	
	Case (*n* = 101)	*p* Value (RNF213)	Case (*n* = 103)	*p* Value (RNF213)	Case (*n* = 515)	*p* Value (*RNF213*)	Case (*n* = 3453)	Control (*n* = 3710)
Age, *n* (100%)		0.357		0.632		0.958		
≤60	41 (40.59%)		51 (49.51%)		159 (30.87%)		1669 (48.33%)	2013 (54.26%)
>60	60 (59.41%)		52 (50.49%)		337 (65.44%)		1784 (51.67%)	1697 (45.74%)
Unknown	0		0		19 (3.69%)		0	0
Gender		0.979		0.176		0.617		
Female	37 (36.63%)		65 (63.11%)		276 (53.59%)			
Male	64 (63.37%)		38 (36.89%)		239 (46.41%)			
Stage, *n* (100%)		0.795		0.011		0.867		
I	54 (53.47%)		51 (49.51%)		276 (53.59%)			
II	29 (28.71%)		17 (16.51%)		121 (23.50%)			
III/IV	18 (17.82%)		35 (33.98%)		110 (21.36%)			
Unknown	0		0		8 (1.55%)			
Pathologic T, *n* (100%)		0.306		0.672		0.581		
T1/T2	89 (88.12%)		85 (82.52%)		446 (86.60%)			
T3/T4	12 (11.88%)		18 (17.48%)		66 (12.82%)			
Unknown	0		0		3 (0.58%)			
Pathologic N, *n* (100%)		0.646		0.020		0.744		
N0	71 (70.30%)		64 (62.14%)		332 (64.47%)			
N1	14 (13.86%)		10 (9.71%)		95 (18.45%)			
N2/N3	16 (15.84%)		29 (28.16%)		76 (14.76%)			
Unknown	0		0		12 (2.30%)			
Pathologic M, *n* (100%)		0.230		0.282		0.757		
M0	83 (82.18)		101 (98.06%)		346 (67.18%)			
M1	1 (0.99%)		2 (1.94%)		25 (4.85%)			
Unknown	17 (16.83%)		0 (0%)		144 (27.77%)			

## Data Availability

The datasets generated during and/or analyzed during the current study are not publicly available but are available from the corresponding author on reasonable request.

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
