# Peer review of "Crosstalk among Alternative Polyadenylation, Genetic Variants and Ubiquitin Modification Contribute to Lung Adenocarcinoma Risk"

_ijms, 2024, doi:10.3390/ijms25158084_

Round 1

Reviewer 1 Report

Comments and Suggestions for Authors

In "Crosstalk among alternative polyadenylation, genetic variants and ubiquitin modification contribute to lung adenocarcinoma risk", Wu and colleagues set out to degine a role for APA-related E3 ubiquitin ligases in lung adenocarcinoma (LUAD). 

They first detect differentially regulated genes in LUAD vs normal tissue using bio-informatic analysis of tumor transcriptome and proteome databases, filter for APA-related genes, and identify 12 such genes with 49 SNP's. Additional database queries identify 5 RNF213 SNP's that are correlated to high risk, and show that RNF213 expression itself is correlated with e.g. poor survival. They additionally show that ZBTB20 binds to RNF213, and is ubiquinated and subsequently degraded through RND213 activity. ZBTB20 expression itself is correlated to good prognosis. Lastly, the authors provide a model for RNF213 SNP function in LUAD progression through miRNA escape and subsequent RNF213 expression increase.

This is a well-executed study on a potentially important pathway in LUAD. The overall significance of this pathway in LUAD - and its importance for disease monitoring, patient stratification, or therapeutic intervention remains a bit elusive to this reviewer, but there are no scientific objections to the set-up and execution of the study. I therefore recommend publication. 

Comments on the Quality of English Language

Minor editing required

Author Response

Comment: In "Crosstalk among alternative polyadenylation, genetic variants and ubiquitin modification contribute to lung adenocarcinoma risk", Wu and colleagues set out to degine a role for APA-related E3 ubiquitin ligases in lung adenocarcinoma (LUAD). 

They first detect differentially regulated genes in LUAD vs normal tissue using bio-informatic analysis of tumor transcriptome and proteome databases, filter for APA-related genes, and identify 12 such genes with 49 SNP's. Additional database queries identify 5 RNF213 SNP's that are correlated to high risk, and show that RNF213 expression itself is correlated with e.g. poor survival. They additionally show that ZBTB20 binds to RNF213, and is ubiquinated and subsequently degraded through RND213 activity. ZBTB20 expression itself is correlated to good prognosis. Lastly, the authors provide a model for RNF213 SNP function in LUAD progression through miRNA escape and subsequent RNF213 expression increase.

This is a well-executed study on a potentially important pathway in LUAD. The overall significance of this pathway in LUAD - and its importance for disease monitoring, patient stratification, or therapeutic intervention remains a bit elusive to this reviewer, but there are no scientific objections to the set-up and execution of the study. I therefore recommend publication. 

Resposne: Dear reviewer, on behalf of all the authors, I extend our heartfelt gratitude and appreciation to you. We sincerely appreciate your professionalism and patience throughout the review process. Your valuable insights and suggestions have been instrumental in refining our manuscript.

We are delighted to receive your approval for publication in your journal. This is a great honor for us and a recognition of our research efforts. We will continue to strive for excellence to ensure the quality and academic value of our work.

Thank you once again for your guidance and support. We look forward to continuing our exchange and sharing of research outcomes in future collaborations.

Reviewer 2 Report

Comments and Suggestions for Authors

Comments: Crosstalk among alternative polyadenylation, genetic variants  and ubiquitin modification contribute to lung adenocarcinoma  risk

This manuscript aims to elucidate the complex interplay between alternative polyadenylation (APA), genetic variants, and ubiquitin modification in the progression of lung adenocarcinoma (LUAD). The authors identify RNF213 as a key E3 ubiquitin ligase associated with LUAD, influenced by specific genetic loci (apaQTL-SNPs) that modulate its expression. The study suggests that RNF213 promotes LUAD by mediating the ubiquitination and degradation of ZBTB20, a potential tumor suppressor. Despite the comprehensive approach and significant findings, several critical issues and methodological concerns need to be addressed before the paper can be considered for publication.

Major Comments:

1. Author attempts to link APA, genetic variants, and ubiquitin modification in LUAD, but the hypothesis and focus are unclear due to extensive background. The introduction should concisely state the hypothesis and research objectives, clearly explaining the interaction between RNF213 and ZBTB20 in LUAD progression. APA’s role should be better connected to the ubiquitin-proteasome system (UPS). Lines 53-57 on RNF213 need restructuring for clarity. I suggest simplifying the background and focusing on key aspects.

2. I recommend including the actual correlation scores, like Pearson’s r values, instead of just stating that there are 1,420 genes with consistent mRNA and protein levels. These scores will provide a clearer and more quantitative picture of how well mRNA and protein levels match up, which is important for understanding the relationship. Studies like those by Mertins et al. (2016) and Zhang et al. (2016) typically find correlation ranges around 0.2 to 0.5. Including this data will strengthen your analysis.

3. Author should provide more details on how you managed batch corrections and data analysis between the two different proteomics methods used as well as integration with gene expression? It's important to explain how you dealt with a potential batch effects that might come from differences in experimental conditions, reagents, instruments, or processing times. Also, please describe how you normalized or adjusted the datasets to make sure they can be compared accurately. Adding this information will make your study more reliable and easier to replicate.

4. The authors need to explain more about how RNF213 affects ZBTB20 to understand how it helps lung cancer grow. They should study the specific pathways affected and show how this interaction makes lung cancer worse. Studying these pathways in detail would give stronger evidence for their idea.

Minor Comments:

1. In the abstract, lung adenocarcinoma (LUAD) appeared without its full abbreviation explained, which could confuse readers unfamiliar with the term.

2. Author should explain why they chose to use log2(FPKM+1) instead of just FPKM in your analysis?

3. Line 328 and 329, Please specify which software you used in your analysis: ITScorePP or ITScorePR? This detail is crucial for understanding your methodology and for the reproducibility of your results.

4. In the method section 4.4 and 4.5, author should club together with title “Cell culture and transfection”.

Comments on the Quality of English Language

Minor editing of English language required!

Reviewer 3 Report

Comments and Suggestions for Authors

The work by Wu Y. et al. titled "Crosstalk among alternative polyadenylation, genetic variants, and ubiquitin modification contribute to lung adenocarcinoma risk" aims to elucidate novel insights into gene regulation and associated biological processes in lung adenocarcinoma (LUAD) by investigating the roles of ubiquitin modification and alternative polyadenylation, identifying APA-related E3 ubiquitin ligases, and examining the interaction and functional mechanisms of RNF213 and ZBTB20.

The work is primarily associative. The multi-step approach for the selection process for APA-related E3s in LUAD is correct. Further analysis of public datasets reveals that RNF123 is more highly expressed in LUAD tumors compared to normal tissues and is positively correlated with poor prognosis. Additionally, ZBTB20 is identified as a potential substrate for ubiquitination by RNF213 in LUAD, still using publicly available datasets. As a minor observation, please note that at line 69, "It is may the heart of the matter" should be deleted.

I find the title somewhat out of context and not fully aligned with the content of the manuscript, which, as noted, relies over 80% and more on associative evaluations derived from public datasets. The work is mainly performed on public databases; only towards the end do the authors demonstrate the function of RNF213 in ZBTB20 ubiquitination and its biological role using an RNF213 knockdown model in the SPCA1 cell line. Experimental data must be added for at least one more cell line.

Unfortunately, the conclusion in the discussion, "In this study, we delineate the role of RNF213 in the progression of LUAD," is not entirely consistent with the data presented, which need to be strengthened. Other conclusions such as "...five apaQTL-SNPs (rs41301932, rs4494603, rs9890400, rs56066320, rs3341123) within a 1 Mb range of RNF213 regulate its expression" are presumptive. It is not merely a matter of semantics, but the level of experimental data that mechanistically demonstrates a phenomenon. Through public database interrogation, only associations can be studied, and hypotheses must then be verified in more complex experimental models. The aforementioned polymorphisms do not "regulate" but "are associated with." In this regard, the authors should moderate their claims.

The limitations of conducting scientific works mainly querying public datasets include the inability to establish causality and the reliance on pre-existing data, which may lack the depth and specificity needed for comprehensive mechanistic insights. While public datasets provide valuable associative data and hypotheses, these must be validated in more complex experimental models to confirm the biological relevance and mechanistic pathways involved.

Comments on the Quality of English Language

The quality of English is good.

Round 2

Reviewer 3 Report

Comments and Suggestions for Authors

I am positively impressed by the serious and diligent effort these authors have invested in addressing my comments promptly and satisfactorily, significantly enhancing the quality and scientific rigor of their manuscript.